# Temporal Stability of Responses to the Edinburgh Postpartum Depression Scale by Bedouin Mothers in Southern Israel

**DOI:** 10.3390/ijerph192113959

**Published:** 2022-10-27

**Authors:** Samira Alfayumi-Zeadna, Norm O’Rourke, Zuya Azbarga, Miron Froimovici, Nihaya Daoud

**Affiliations:** 1Center for Women’s Health Studies and Promotion, Ben-Gurion University of the Negev, Be’er Sheva 84417, Israel; 2Nursing Department, School of Health Sciences, Ashkelon Academic College, Ashkelon 78682, Israel; 3MAP Centre for UrbanHealth Solutions, Li Ka Shing Knowledge Institute, Michael’s Hospital, Unity Health Toronto, Toronto, ON M5B 1W8, Canada; 4Multidisciplinary Center for Research on Aging, Ben-Gurion University of the Negev, Be’er Sheva 84417, Israel; 5Department of Psychology, Ben-Gurion University of the Negev, Be’er Sheva 84417, Israel; 6School of Public Health, Faculty of Health Sciences, Ben-Gurion University of the Negev, Be’er Sheva 84417, Israel; 7Clalit Health Services, Southern Region, Be’er-Sheva 84417, Israel

**Keywords:** Bedouin, depression, Edinburgh Postpartum Depression Scale, pregnancy, postpartum

## Abstract

The detection and treatment of mental illness during pregnancy and postpartum are essential for the well-being of both mother and child. For this study, we set out to estimate the prevalence of perinatal depression among Bedouin mothers in southern Israel and determine if the latent structure of responses to the Edinburgh Postpartum Depression Scale (EPDS)—including depression, anxiety, and anhedonia—previously reported also applies to this population. A total of 332 women completed the EPDS during pregnancy (26–38 weeks) and again 2–4 months postpartum. Confirmatory factor analyses were performed to determine if first-order factors were correlated, or instead measure a second- or higher-order latent construct. We next performed temporal invariance analyses to compare the latent structure of EPDS responses over time. When pregnant, 35% of women provided EPDS responses suggestive of elevated depressive symptomology; this decreased to 23% postpartum. At both points, each EPDS factor significantly measures a higher-order, latent construct. The EPDS appears to measure three factors, labeled sadness, anxiety, and anhedonia. This latent structure appears stable (i.e., during pregnancy and postpartum). Further research is needed to validate EPDS responses versus structured clinical interviews. The construct validity of EPDS factors should be examined across other at-risk groups and over time.

## 1. Introduction

Depression and anxiety are common during pregnancy and the first year postpartum, with significant adverse effects on maternal mortality [1], attachment, neonatal health, and early-life development [2]. These adverse effects have short- and long-term consequences on mothers’ health and infants’ cognitive and emotional development [3,4]. Ten to twenty percent of pregnant and postpartum women experience clinical depression and anxiety [5]. Depressive symptoms reported during pregnancy are likely to persist postpartum [5], with only a slight improvement over time [6]. Symptoms include sadness, hopelessness, fatigue, inactivity, sleep disturbances, uncontrollable crying, suicidal ideation, and obsessive thoughts of harming their newborn [7]. Depression during pregnancy and postpartum is associated with anxiety, stressful life events, low education, poor economic status, a history of mental illness, a lack of social support, domestic violence, unplanned pregnancy, prior miscarriages, and poor infant health [8,9,10,11].

### 1.1. Perinatal Depression and Disadvantaged Minorities

Depression during pregnancy and postpartum is more prevalent among disadvantaged groups and ethnic minorities [10,12]. Minority women in Israel and abroad live with fewer socioeconomic resources and experience more life stressors [13]. Moreover, these women commonly face multiple barriers to treatment, which also increase depression risk. These include limited awareness of perinatal depression by women and their families, insufficient social support, and inadequate detection and treatment at healthcare services [14].

### 1.2. Bedouin in Israel

Israel is a multicultural nation of fewer than 10 million people, comprised of a Jewish majority and a large (21.1%) Arab minority [15]. Within Arab society, the Bedouin are a unique social and ethnonational minority who comprise one-third of the population in the southern Negev region [16]. The Bedouin are a traditional, Muslim, pastoral people, characterized by unique customs and collective identity. These include extended family structures, polygamy, and consanguineous marriages (e.g., first cousins) [17]. Additionally, fertility rates are higher among the Bedouin in southern Israel (5.5) compared to other Arab women and Jewish Israelis (both 3.1) [16]. Preterm delivery is also high, at 24% [18].

The Bedouin live in both the north and south of Israel, Jordan, Syria, and across North Africa. In Israel, a growing proportion of Bedouin today live in cities and towns, but many still live in semi-permanent villages (both recognized and legally unrecognized) with limited or no utilities such as electricity, running water, public transportation, or access to health services [19].

The Bedouin are also economically disadvantaged, with low education and socioeconomic status and high unemployment compared to other Israelis; this subpopulation is the poorest in Israel [20]. Arab-Israelis comprise 16.9% of students in higher education [21], but in 2013, only 8.6% of Bedouin women in southern Israel had academic degrees [22]. Only 21% of Bedouin women in southern Israel are in the paid workforce compared to more than a third of all Arab women (36%) and 83% of Jewish women in Israel [23].

It is estimated that postpartum depression is higher among Arab than Jewish women in Israel (20.8% vs. 7%, respectively) [24]. During the recent COVID-19 pandemic, this may have more than doubled for Arab women [10]. The rates of perinatal depression among Bedouin women are also significantly larger than the general population, with estimates ranging from 31% to 58% [10,11,13]. Sociodemographic and cultural factors such as a lack of partner support, low income, low education, and unplanned pregnancy are significant predictors of postpartum depression among Bedouin women in Israel [11].

In addition, Bedouin women face various barriers that may impede mental health detection and treatment; these include limited mental health services, unreliable transportation, limited knowledge of postpartum depression symptoms and treatment, a lack of culturally appropriate healthcare services, economic barriers and poverty, and the stigmatization of mental disorders [14].

### 1.3. Depression Screening and Detection

Screening, early detection, and interventions for depression are integral to perinatal and maternal mental health. In 2018, the American College of Obstetricians and Gynecologists recommended that care providers screen for both depression and anxiety at least once during the perinatal period; this should include standardized, validated tools such as the Edinburgh Postpartum Depression Scale [25]. They emphasized the importance of early identification to limit negative consequences for mothers and children [26]. Recently, The US Preventive Services Task Force (USPSTF) issued updated clinical guidelines to identify those at risk early in pregnancy. Recommendations include the mental health screening of pregnant and postpartum women, with systems in place to ensure accurate diagnoses, treatment, and follow-up care [27].

### 1.4. Edinburgh Postpartum Depression Scale

The EPDS was developed to screen for depression during pregnancy and postpartum [25,28]. Responses of 11/12+ to the original English EPDS correspond to 86% sensitivity and 78% specificity compared to blind clinical ratings of clinical depression [25,29]. Cox et al. (1987) reported that a score of 12/13 was found to identify women with major depression [25]. Yet, the false-negative rate was high. Thus, they recommended a cut-off score of 9/10 to heighten detection [25]. Moreover, Levis et al. (2020) reported that an EPDS cut-off value of 11 or higher optimized sensitivity and specificity. However, lower cut-off values could be used if the intention is to avoid false negatives and identify most patients who meet diagnostic criteria [28].

The EPDS is a commonly used perinatal depression screening measure [28]. Subsequent research has identified a multi-dimensional structure of responses. In their factor analysis, Bina and Harrington (2016) reported that the Hebrew version of the EPDS measures two main dimensions, anxiety (three items) and depression (six items) [30]. However, a population-based sample using the English version of the EPDS identified a three-factor structure (i.e., anxiety, depression, and anhedonia), stable over four time points; namely, 18 and 32 weeks’ gestation and 8 weeks and 8 months postpartum [31]. Like Bina and Harrington (2016), a majority of studies indicate that depression and anxiety are the main factors, irrespective of language and culture [31]. Others suggest a third factor, such as self-harm (English [32]). These differences may be attributable to different population characteristics and times of administration.

For the current study, we set out to estimate the prevalence of perinatal depression among Bedouin women in southern Israel, while pregnant, and again 2–4 weeks postpartum. We also set out to determine if the three-factor structure (depression, anxiety, anhedonia) of EPDS responses reported with other populations applies to Bedouin women. Completing the EPDS during pregnancy and 2–4 months postpartum allowed us to test the temporal consistency of the latent structure and EPDS item responses.

## 2. Materials and Methods

### 2.1. Sample and Ethics

Bedouin women in southern Israel were recruited at women’s health clinics to take part in a longitudinal, non-randomized, controlled trial [32]. This study received ethical approval from the Helsinki Committee of the Clalit Health Services (COM-004-16). All participants were Bedouin women who were 26 to 38 weeks pregnant. This sample included only single-birth pregnancies. The study included 332 of the 384 eligible women recruited during pregnancy at baseline. A total of 332 women participated in two face-to-face interviews, in which the EPDS was twice administered: at 26–38 weeks of pregnancy and 2–4 months postpartum.

### 2.2. Study Procedures and Data Collection

Data collection was conducted in two women’s health centers from October 2017 to February 2018. Bedouin women receiving medical services at one of two women’s health centers were approached by a physician or nurse who requested their participation. Those who agreed and met inclusion criteria received study information in Arabic that described the study, inclusion criteria, researchers, email contact information, and the consent form. Participants were interviewed face to face using a structured questionnaire administered by a female nurse or research assistant in a separate clinic room. The participation rate was 82%.

Healthcare is universal in Israel, supported by monthly premiums and progressive taxation. All residents are insured by one of four healthcare organizations. This includes full perinatal care for all Arab and Jewish citizens. In other words, (almost) all women attend these clinics, including the Negev Bedouin.

### 2.3. Instrument

#### Edinburgh Postpartum Depression Scale (EPDS)

The EPDS is a 10-item, self-reported scale with responses reported on a Likert scale ranging from 0 to 3 with a maximum total score of 30 and higher scores representing greater symptom severity. The EPDS was developed and validated to detect depression during the pregnancy and postpartum period [25]. We used the Arabic version of the EPDS [33], which has been used widely in the Middle East, including with Bedouin and Arab women in Israel. The internal consistency of EPDS responses by Arab women in Israel is high (α = 0.84) [10,11,24]. In addition, the sensitivity and specificity for the Arabic version of the EPDS have been reported as 91% and 84%, respectively [33].

Using the standard EPDS cut-off of 10+, depression prevalence may be as high as 34% among Bedouin mothers but lower at 26% when a 13+ cut-off is applied [11,13]. The internal consistency of responses to the Arabic version of the EPDS by Bedouin mothers was high (α = 0.80) [13]. In the current analysis, we used the three factors of EPDS: anhedonia (2 items: 1 and 2), anxiety (4 items: 3, 4, 5, and 6), and depression (4 items: 7, 8, 9, and 10).

### 2.4. Statistical Procedures

Confirmatory factor analyses (CFA) were first performed to replicate the 3-factor model of EPDS responses, as reported with other populations (e.g., [34]). We also set out to determine if first-order factors were correlated or mapped onto a second- or higher-order latent construct [35].

Assuming that the 3-factor model fits effectively during pregnancy and postpartum (nested models), our intent was to perform temporal invariance analyses to compare the latent structure of EPDS responses over time. This allows us to ascertain if the factor structure is consistent over time or if specific facets of depressive symptomology differ before and after birth. Temporal invariance analyses are a psychometric extension of test–test reliability, unaffected by natural variability in symptom levels (e.g., [36]). Previous research has established the consistency of EPDS factor structure during pregnancy and after birth [34]; this is the first study, however, to examine the temporal consistency of the EPDS latent structure and item responses over time.

Three goodness-of-fit indices are reported to assess goodness of fit: An incremental, an absolute, and a parsimonious fit index. The comparative fit index (CFI) is an incremental index representing the extent to which a hypothesized model is a better fit to data than the null model. Coefficient values greater than 0.94 for the CFI indicate a good model fit [37]. The standardized root mean square residual (SRMR) is an *absolute index* that represents the standardized difference between observed and predicted correlations within a hypothesized model. Finally, the root mean square error of approximation (RMSEA) is a *parsimony index* that represents the extent to which a hypothesized model fits the data relative to the general population. Coefficient values less than 0.055 for the SRMR and RMSEA suggest a good model fit [38]. CFA and invariance analyses were performed using AMOS 26.0 (IBM: New York, NY, USA) and the maximum likelihood method of parameter estimation [37].

## 3. Results

For this study, 332 Bedouin women completed the EPDS during pregnancy and postpartum. Responses were moderately correlated: r = 0.40, *p* < 0.01.

The mean age of participants was 28.5 years (SD = 6.1); all of them were married; 31.6% had 2–3 and 25.3% had 4+ children during pregnancy; gestational age was 29 weeks (SD = 3.0); 45% were in a consanguineous marriage; 35% experienced miscarriages; 13.3% lived in a polygamous marriage; only 14.8% had an academic degree; 82.8% were unemployed; and 57% had an income below the average in the country. See Table 1.

During pregnancy, 35.2% of women provided EPDS responses of 10+, suggestive of clinically significant symptomology; this dropped to 23.2% postpartum. See Table 2.

Confirmatory Factor Analysis (CFA). We performed CFA to replicate the three-factor structure of EPDS responses with Bedouin women and to determine if the factors are just correlated (first-order model) or if they map onto a second- or higher-order latent construct. Both first- and higher-order models fit the data (χ^2^ = 38.09, *df* = 26, *p* < 0.06; χ^2^ = 30.90, *df* = 26, *p* < 0.23, respectively), but the smaller χ^2^ value supports the higher-order model (i.e., ∆χ^2^ = 7.19, *p* < 0.01). Moreover, the goodness of fit statistics are superior for the SRMR (i.e., 0.034 vs. 0.039) and RMSEA (i.e., 0.025 vs. 0.037); the CFI is identical (i.e., 0.99).

The results suggest that EPDS responses by Bedouin women are effectively measured by a higher-order model, with three first-order factors mapping onto a higher-order depression latent construct. This finding emerged for EPDS responses during pregnancy and postpartum.

When reported while pregnant, the responses by Bedouin mothers support this higher-order model: χ^2^ = 30.90, *df* = 26, *p* = 0.23. After correcting for correlated errors between 7 of 55 possible item pairs, the CFI (0.99), SRMR (0.034), RMSEA (0.024), and the full 90 confidence interval for the RMSEA statistic were each within ideal parameters (0 < RMSEA CL_90_ < 0.052). See Figure 1.

Similar results emerged when reported postpartum: χ^2^ = 29.59, *df* = 26, *p* = 0.29. That is, the CFI (0.99), SRMR (0.031), RMSEA (0.020), and the full 90 confidence interval for the RMSEA were again within ideal parameters (0 < RMSEA CL_90_ < 0.050). Statistical power for both higher-order models was estimated at *d* = 0.81 (*n* = 332, *df* = 26), sufficient to detect medium to small effects (where α = 0.05) [38].

Temporal Invariance Analyses. The same factor structure emerged during pregnancy and post-birth; in effect, the same EPDS model was ‘nested’ within mothers (i.e., repeated measures), allowing us to compare the latent structure and item responses over time. The higher-order model of responses fits effectively at both points of measurement, χ^2^ = 60.47, *df* = 52, *p* = 0.20. Each of the three factors significantly measures a higher-order construct.

Yet, the core sadness factors appear to contribute significantly more during pregnancy than postpartum (i.e., β = 0.95 vs. β = 0.73, ∆χ^2^ = 5.29, *p* < 0.05). Women report more difficulties sleeping (β = 0.60 vs. β = 0.50, ∆χ^2^ = 4.66, *p* < 0.05) and crying more often while pregnant (β = 0.85 vs. β = 0.81, ∆χ^2^ = 7.15, *p* < 0.05). Table 3.

## 4. Discussion

We set out to estimate the prevalence of perinatal depression among Bedouin mothers in southern Israel and determine if the latent structure of responses to EPDS—including depression, anxiety, and anhedonia—previously reported also applies to this population. Our results suggest that when pregnant, 35% of women provided EPDS responses suggestive of elevated depressive symptomology; this decreased to 23% postpartum [32]. The results of this study support the three-factor structure of EPDS responses by Bedouin women in Israel at two time points: 26–38 weeks of pregnancy and 2–4 months postpartum. These three factors map onto a higher-order depression latent construct at both time points; specifically, for ‘anxiety’ (items 3, 4, 5, and 6), ‘depression’ (items 7, 8, 9, and 10), and ‘anhedonia’ (items 1 and 2).

This result emerged during pregnancy and postpartum and appears consistent over time despite a significant intermediary event and biological changes during pregnancy and postpartum (i.e., the birth of the child). Our findings are consistent with the results reported by Coates et al. (2017), who examined two antenatal and two postpartum periods [31], and by Matsumura et al. (2020), who examined two postpartum periods at 1- and 6-months postpartum [35]. Consistent with previously published research, the internal reliability of EPDS responses was adequate but not ideal during pregnancy and postpartum (i.e., α = 0.75 and α = 0.77, respectively), and this study only adopted nine out of ten items [35].

With this sample, the EPDS also appears to capture variability between pregnancy and postpartum. For Bedouin women experiencing symptoms of depression, the core sadness factor appears more pronounced during pregnancy; largely, it appears, due to difficulties sleeping and more frequent crying. Other items appear to differ (e.g., anxious/worried, scared/panicky), but the differences are not interpreted, as the overall contribution to the measurement of the anxiety factor is invariant or consistent over time.

Sleeping difficulties and crying can be explained by the physiological effects of pregnancy. Particularly in the third trimester, sleeping can be difficult, and hormonal changes due to pregnancy affect emotional variability [39]. Researchers have reported that perceived total sleep time slightly decreases during the second trimester, followed by a substantial decrease during late pregnancy, and tends to gradually improve during the postpartum period [40]. This difference between points of measurement does not suggest a psychometric deficiency; instead, the EPDS appears to consistently measure a higher-order, three-factor model, with elevated responses to the core factor elevated during pregnancy, which can be ascribed to physiological factors.

A similar three-factor, higher-order model of responses has emerged with other depression screening measures (e.g., Center for Epidemiological Studies—Depression Scale); see [41]. That is, depression appears to have various facets that each measure a higher-order latent construct. Though depression as a diagnostic category is dichotomous, depression as a phenomenon appears multifaceted [35]. The results of this study support that assertion.

A strength of this study was our ability to examine the factor structure of the EPDS over time (i.e., during pregnancy and postpartum) using all 10 items of the EPDS. Some authors have examined the factor structure of the EPDS during pregnancy and postpartum without asking about thoughts of self-harm [31]. Though the frequency of endorsement is generally low for this item, suicide is a leading cause of maternal mortality [42]. We contend that item 10 must be included.

The current study has various limitations. For instance, participants were recruited from two women’s health clinics, which may limit generalizability of findings. In addition, the research took place among women who receive medical services from the same HMO. Further research is required to replicate the factor structure of the EPDS with other disadvantaged groups and ethnic minorities. We were able to estimate the point prevalence of depression, but these percentages are estimates only. Future research is needed to corroborate EPDS response levels vis-à-vis comprehensive diagnostic interviews and full mental health assessments. This will help determine which EPDS cut-off value is most appropriate for this population.

## 5. Conclusions

In conclusion, the EPDS appears to measure three factors (sadness, anxiety, and anhedonia) with a stable, higher-order factor structure measured during pregnancy and postpartum. Our findings revealed high goodness-of-fit and measurement invariance or consistency over several months. These findings suggest that the EPDS can continue to be used to screen for elevated levels of depressive symptoms in Bedouin women.

Brief, valid instruments for the assessment of mental health during pregnancy and postpartum are needed in community healthcare serving pregnant women and new mothers. Further research is needed to corroborate the names currently assigned to the three EPDS factors. Other factor labels may better capture the essence of each construct. It may also be advantageous to include additional items such as tokophobia (fear of childbirth), which is often high, heightens anxiety, and increases the risk of C-section [43]. Other items might measure posttraumatic stress, bonding difficulties, and intrusive fantasies of harm to the infant. Ultimately, improved measurement will enable better detection and targeted intervention programs that foster early child development and the well-being of the mother and the family. We also suggest preventive interventions, screening, and treatment (medication and psychotherapy) in early pregnancy to improve social support and parenting skills. Perinatal education is also required, with follow-up, to increase healthcare access and knowledge about postpartum depression, especially among women at risk and in disadvantaged populations.

## Figures and Tables

**Figure 1 ijerph-19-13959-f001:**
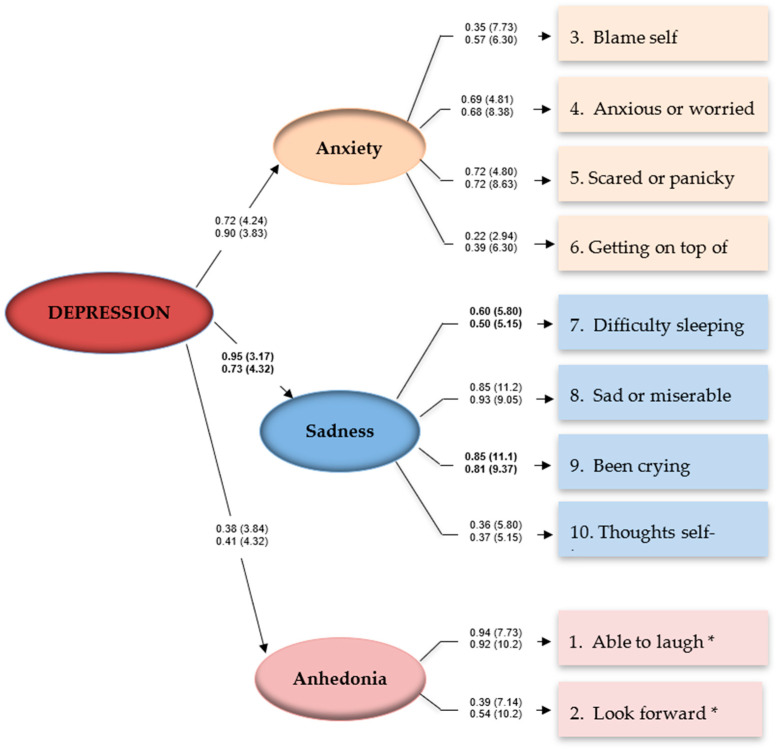
Three-factor model of depressive symptoms, pregnancy and postpartum, Bedouin mothers in Southern Israel. Note: Coefficients for the pregnancy model are above coefficients for the post-birth model. Parameters are expressed as maximum likelihood estimates (standardized solutions). Parenthetical numbers indicate significance levels (CR > 1.96, *p* < 0.05; CR > 2.58, *p* < 0.01). * Anhedonia items are reverse-coded.

**Table 1 ijerph-19-13959-t001:** Sociodemographic and clinical characteristics of participants (N = 332).

Variable	N (%)	Mean (SD)	Range
Maternal Age		28.5 (6.1)	15–42
15–24	93 (28.0)		
25–34	179 (53.9)		
≥35	60 (18.1)		
Number of children			0–11
0–1	143 (43.1)		
2–3	105 (31.6)		
≥4	84 (25.3)		
Polygamous marriages			
No	288 (86.7)		
Yes	44 (13.3)		
Education			
Academic degree	49 (14.8)		
Non-academic degree	238 (85.2)		
Employment status			
Employed	57 (17.2)		
Unemployed	275 (82.8)		
Family income (ILS/month)			
≥15,149	141 (42.5)		
<15,149	191 (57.5)		
Consanguineous marriage			
Yes	149 (45%)		
No	182 (55%)		
Experienced miscarriages			
No	220 (65%)		
Yes	112 (35%)		
Gestational age (weeks)		29 (3.0)	26–38

**Table 2 ijerph-19-13959-t002:** Edinburgh Depression Scale descriptive statistics, pregnancy and post-birth.

Variable	Mean	SD	Skewness	Kurtosis	Alpha (α)
EPDS—Pregnancy	8.07	5.29	0.80	0.31	0.75
Anhedonia T1	1.88	1.67	0.60	−0.42	
Anxiety T1	3.87	2.56	0.46	0.98	
Sadness T1	2.32	2.66	1.24	1.51	
EPDS—Post Birth	6.53	4.75	0.83	0.83	0.77
Anhedonia T2	1.50	1.54	0.93	0.31	
Anxiety T2	3.22	2.64	0.74	−0.09	
Sadness T2	1.80	2.09	1.39	1.51	

**Table 3 ijerph-19-13959-t003:** Invariance analyses of depressive symptoms—pregnancy and post-birth, Bedouin women in Israel.

Comparison	χ^2^	Δχ^2^	*df*	Δ*df*	SRMR	CFI	RMSEA (90_CI_)
baseline	60.473	-	52	-	0.034	0.99	0.016 (0–0.030)
Depression—anxiety	61.937	1.464	53	1	0.033	0.99	0.016 (0–0.031)
Depression—sadness	67.230	5.293 *	54	1	0.039	0.99	0.019 (0–0.033)
Depression—anhedonia	67.230	0.096	54	1	0.039	0.99	0.019 (0–0.033)
Anxiety							
EPDS03	69.181	1.951	55	1	0.041	0.99	0.020 (0–0.033)
EPDS04	74.762	5.581	56	1	0.041	0.99	0.022 (0.002–0.035)
EPDS05	78.920	4.158	57	1	0.042	0.99	0.024 (0.008–0.036)
EPDS06	78.920	2.270	57	1	0.042	0.99	0.024 (0.008–0.036)
Sadness							
EPDS07	83.582	4.662 *	58	1	0.044	0.99	0.026 (0.012–0.037)
EPDS08	85.463	1.881	59	1	0.045	0.99	0.026 (0.012–0.038)
EPDS09	92.615	7.152 **	60	1	0.045	0.98	0.029 (0.016–0.040)
EPDS10	92.615	0.031	60	1	0.045	0.98	0.029 (0.016–0.040)
Anhedonia							
EPDS01	95.423	2.807	61	1	0.045	0.98	0.028 (0.016–0.039)
EPDS02	95.423	3.737	61	1	0.043	0.98	0.028 (0.016–0.039)

* *p* < 0.05; ** *p* < 0.01. *Note. df* = degrees of freedom; SRMR = standardized root mean square residual; CFI = comparative fit index; RMSEA = root mean square error of approximation; RMSEA CL_90_ = 90% confidence limits for the RMSEA statistic.

## Data Availability

Anonymized data are available from the corresponding author upon reasonable request.

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
