# Peer review of "Temporal Stability of Responses to the Edinburgh Postpartum Depression Scale by Bedouin Mothers in Southern Israel"

_ijerph, 2022, doi:10.3390/ijerph192113959_

Round 1
Reviewer 1 Report
I would like to thank the authors for the opportunity to read the article. Below are some comments for the authors to consider
Title
It is not necessary to mention the Edinburgh Postpartum Depression tool in the title. Enough: depression.
1. Introduction
The authors listed the most important factors determining the development of depression in pregnant women.
Bedouin in Israel
Can the authors provide more accurate data on the education and material status of Bedouins compared to the rest of the Israeli population?
Depression Screening and Detection
What other tools for measuring depression in pregnant women could be complementary to this tool?
2. Results
How representative is the group of Bedouin women in terms of education, employment status and income?
3. Discussion
When starting the discussion of the results, it is good to summarize the previous findings and indicate the purpose of the research.
Limitations.
It should be indicated that some Bedouin women do not use a doctor and perhaps these results would be different in a larger population.
Author Response
Dear reviewer,
Thank you so much for your helpful feedback, comments, and suggestions, which helped to improve the quality, readability, and impact of our manuscript. We hope that you will find this revised manuscript suitable for publication.
|
Reviewers’ Comments |
Responses |
|
Reviewer #1 |
|
|
Comment 1: Title: It is not necessary to mention the Edinburgh Postpartum Depression tool in the title. Enough: depression. |
EPDS is the main topic in this study, and “Temporal stability” in the title refers to This scale. Note: The word “scale” was missing from the title, we added it. |
|
Comment 2: Introduction: The authors listed the most important factors determining the development of depression in pregnant women. Bedouin in Israel. Can the authors provide more accurate data on the education and material status of Bedouins compared to the rest of the Israeli population? |
We added in the introduction more data regarding the education and material status of Bedouin compared to other Israelis. Page 2, lines 64-74. |
|
Comment 3: Depression Screening and Detection: What other tools for measuring depression in pregnant women could be complementary to this tool?
|
The EPDS is a commonly used perinatal depression screening measure. And it’s mandatory for screening in all maternal and child health clinics and women’s health centers in Israel. We used the Arabic version of the EPDS which has been used widely in the Middle East including Bedouin and Arab women in Israel. Sensitivity and specificity for the Arabic version of the EPDS have been reported as 91% and 84%, respectively. (Page 3, 140-144). These findings suggest that administration of additional depression screening measures is not required; the EPDS is sufficient as a screening measure. |
|
Comment 4: Results How representative is the group of Bedouin women in terms of education, employment status and income?
|
Healthcare is universal in Israel, supported by monthly premiums and progressive taxation. All residents are insured by one of four healthcare organizations. This includes full perinatal care for all Arab and Jewish citizens. In other words, (almost) all women attend these clinics, including the Negev Bedouin. There is no selection bias. We have added this important point to the text. Page 4, lines 144-147. |
|
Comment 5: Discussion: When starting the discussion of the results, it is good to summarize the previous findings and indicate the purpose of the research. |
We indicated the purpose of the study and added a summary of the findings in at the beginning of the discussion. Page 7, lines 242-246.
|
|
Comment 6: Limitations It should be indicated that some Bedouin women do not use a doctor and perhaps these results would be different in a larger population.
|
The two clinics from which women were recruited are specific to the Bedouin population (i.e., almost all patients are Bedouin women). That is, clinics are located in Arab sector to serve the Bedouin community. Few women today forego healthcare. |
Reviewer 2 Report
The studies are very interesting because they reveal the problems of cultural preparation. But it is precisely for this reason that I feel a significant hunger:1. In my opinion firstly the introduction requires a broader approach to the specificity of the culture and tradition of this minority, due to the fact that western readers might be not familiar with. 2. rates of depression among Jewish then Bedouin women living in Israel are lower due to cultural differences, so living conditions should be presented in more detailed way. 3. Methods: pre and post test of depression symptoms - the statistical analysis is carried out correctly and proves the stability of depressive symptoms during pregnancy and in the postpartum period. 4. Discussion - secondly it is important to be expressed why women with pregnancy depression who received the medical support did not receive any antidepresive treatment that would protect them from the persistence of the disease after childbirth requires clarification. 5. In my opinion, the results and analysis can be extended to reveal the influence of sociological variables on the level of depression: e.g. type of marriage and / or previous miscarriages, as well as the number of children. The authors have this data, so additional statistic analysis will be possible. It would make our knowledge wilder.
6. further research suggestion - An interesting variable related to depression would be the possible disability of the child (probably inconsanguineous marriage this problem is much stronger than in the general population), as I understand it was not controlled, so it may be variable of the planned further research
Author Response
Manuscript (Ref: ijerph-1950254): “Temporal stability of responses to the Edinburgh Postpartum Depression Scale, by Bedouin mothers in southern Israel”.
Dear reviewer,
Thank you so much for your helpful feedback, comments, and suggestions, which helped to improve the quality, readability, and impact of our manuscript. We hope that you will find this revised manuscript suitable for publication.
|
Reviewers’ Comments |
Response |
|
Reviewer #2 |
|
|
Comment 1: In my opinion, firstly the introduction requires a broader approach to the specificity of the culture and tradition of this minority, due to the fact that western readers might be not familiar with.
|
We added more information regarding Bedouin minority population in the introduction. Page 2, lines 58-73. |
|
Comment 2: Rates of depression among Jewish then Bedouin women living in Israel are lower due to cultural differences, so living conditions should be presented in more detailed way.
|
We added more information regarding sociodemographic and cultural risk factors of postpartum depression and barriers to screening and treatment among Bedouin women. Page 2, lines 79-86 |
|
Comment 3:
Methods: pre and post test of depression symptoms - the statistical analysis is carried out correctly and proves the stability of depressive symptoms during pregnancy and in the postpartum period.
|
Thank you! |
|
Comment 4: Discussion - secondly it is important to be expressed why women with pregnancy depression who received medical support did not receive any antidepressive treatment that would protect them from the persistence of the disease after childbirth requires clarification. |
We emphasized the importance of early detection and treatment to prevent the persistence of depression after childbirth. In the discussion, page 9, lines 318-322. |
|
Comment 5: In my opinion, the results and analysis can be extended to reveal the influence of sociological variables on the level of depression: e.g. type of marriage and / or previous miscarriages, as well as the number of children. The authors have this data, so additional statistic analysis will be possible. It would make our knowledge wilder.
|
This study is part of a longitudinal program of research, including a non-randomized controlled trial (Reference 31), which examined the influence of these sociological data on the level of depression. See please: 10.1007/s10995-022-03434-1 ( 2022 Apr 21) The main aim of this study is to determine if the latent structure of responses to the Edinburgh Postpartum Depression Scale (EPDS) including depression, anxiety, and anhedonia, applies to this population during pregnancy and postpartum. . |
|
Comment 6: further research suggestion - An interesting variable related to depression would be the possible disability of the child (probably consanguineous marriage this problem is much stronger than in the general population), as I understand it was not controlled, so it may be variable of the planned further research
|
In this study and in a recent published study (reference 12 : DOI: 10.1002/da.22290), we didn’t find a relationship between consanguineous marriage and postpartum depression symptoms. Moreover, in a previous study among Bedouin women (Alfayumi-Zeadna S., 2015), we found that consanguineous marriage was a protective factor. This may indicate that in Bedouin culture, which favors marriage within the extended family, consanguineous marriage provides structural protection and support for mothers. The strongest held belief for those who married within the clan was that consanguinity promotes stability, the preservation of traditions, and the continuity of a culture and a way of life. The reviewer, of course, is correct that consanguineous marriage increases the risk for various diseases and disabilities.
|
Round 2
Reviewer 2 Report
Dear Authors and Editors,
Thank you for giving me the oportunity to read very interesting article and get more aware of Bedouin women living conditions and problems.
The article after revision is accepted for publication.
Congratulations,
Joanna Kossewska